# Accurate, reliable and fast robustness evaluation

**Wieland Brendel**[1,3]    **Jonas Rauber**[1-3]    **Matthias Kümmerer**[1-3]    **Ivan Ustyuzhaninov**[1-3]

**Matthias Bethge**[1,3,4]

[1]Centre for Integrative Neuroscience, University of Tübingen
[2]International Max Planck Research School for Intelligent Systems
[3]Bernstein Center for Computational Neuroscience Tübingen
[4]Max Planck Institute for Biological Cybernetics
`wieland.brendel@bethgelab.org`

## Abstract

Throughout the past five years, the susceptibility of neural networks to minimal adversarial perturbations has moved from a peculiar phenomenon to a core issue in Deep Learning. Despite much attention, however, progress towards more robust models is significantly impaired by the difficulty of evaluating the robustness of neural network models. Today's methods are either fast but brittle (gradient-based attacks), or they are fairly reliable but slow (score- and decision-based attacks). We here develop a new set of gradient-based adversarial attacks which (a) are more reliable in the face of gradient-masking than other gradient-based attacks, (b) perform better and are more query efficient than current state-of-the-art gradient-based attacks, (c) can be flexibly adapted to a wide range of adversarial criteria and (d) require virtually no hyperparameter tuning. These findings are carefully validated across a diverse set of six different models and hold for $L_0$, $L_1$, $L_2$ and $L_\infty$ in both targeted as well as untargeted scenarios. Implementations will soon be available in all major toolboxes (Foolbox, CleverHans and ART). We hope that this class of attacks will make robustness evaluations easier and more reliable, thus contributing to more signal in the search for more robust machine learning models.

## 1   Introduction

Manipulating just a few pixels in an input can easily derail the predictions of a deep neural network (DNN). This susceptibility threatens deployed machine learning models and highlights a gap between human and machine perception. This phenomenon has been intensely studied since its discovery in Deep Learning [Szegedy et al., 2014] but progress has been slow [Athalye et al., 2018a].

One core issue behind this lack of progress is the shortage of tools to reliably evaluate the robustness of machine learning models. Almost all published defenses against adversarial perturbations have later been found to be ineffective [Athalye et al., 2018a]: the models just appeared robust on the surface because standard adversarial attacks failed to find the true minimal adversarial perturbations against them. State-of-the-art attacks like PGD [Madry et al., 2018] or C&W [Carlini and Wagner, 2016] may fail for a number of reasons, ranging from (1) suboptimal hyperparameters over (2) an insufficient number of optimization steps to (3) masking of the backpropagated gradients.

In this paper, we adopt ideas from the decision-based boundary attack [Brendel et al., 2018] and combine them with gradient-based estimates of the boundary. The resulting class of gradient-based attacks surpasses current state-of-the-art methods in terms of attack success, query efficiency and

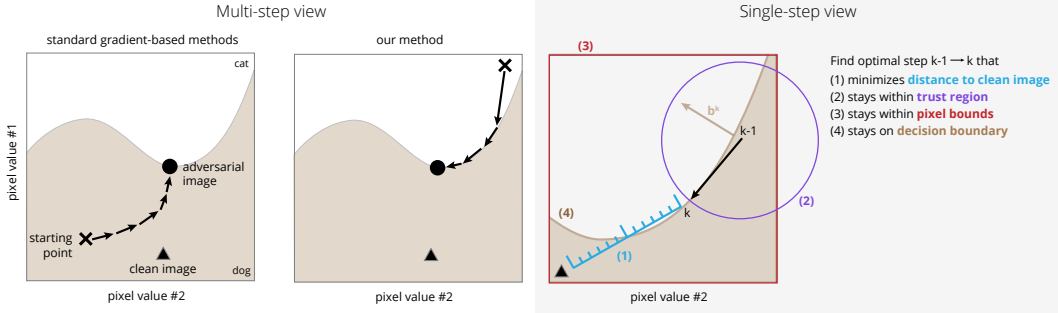

Figure 1: Schematic of our approach. Consider a two-pixel input which a model either interprets as a *dog* (shaded region) or as a *cat* (white region). Given a clean dog image (solid triangle), we search for the closest image classified as a cat. Standard gradient-based attacks start somewhere near the clean image and perform gradient descent towards the boundary (left). Our attacks start from an adversarial image far away from the clean image and walk along the boundary towards the closest adversarial (middle). In each step, we solve an optimization problem to find the optimal descent direction along the boundary that stays within the valid pixel bounds and the trust region (right).

reliability. Like the decision-based boundary attack, but unlike existing gradient-based attacks, our attacks start from a point far away from the clean input and follow the boundary between the adversarial and non-adversarial region towards the clean input, Figure 1 (middle). This approach has several advantages: first, we always stay close to the decision boundary of the model, the most likely region to feature reliable gradient information. Second, instead of minimizing some surrogate loss (e.g. a weighted combination of the cross-entropy and the distance loss), we can formulate a clean quadratic optimization problem. Its solution relies on the local plane of the boundary to estimate the optimal step towards the clean input under the given $L_p$ norm and the pixel bounds, see Figure 1 (right). Third, because we always stay close to the boundary, our method features only a single hyperparameter (the trust region) but no other trade-off parameters as in C&W or a fixed $L_p$ norm ball as in PGD. We tested our attacks against the current state-of-the-art in the $L_0, L_1, L_2$ and $L_\infty$ metric on two conditions (targeted and untargeted) on six different models (of which all but the vanilla ResNet-50 are defended) across three different data sets. To make all comparisons as fair as possible, we conducted a large-scale hyperparameter tuning for each attack. In almost all cases tested, we find that our attacks outperform the current state-of-the-art in terms of attack success, query efficiency and robustness to suboptimal hyperparameter settings. We hope that these improvements will facilitate progress towards robust machine learning models.

## 2   Related work

Gradient-based attacks are the most widely used tools to evaluate model robustness due to their efficiency and success rate relative to other classes of attacks with less model information (like decision-based, score-based or transfer-based attacks, see [Brendel et al., 2018]). This class includes many of the best-known attacks such as L-BFGS [Szegedy et al., 2014], FGSM [Goodfellow et al., 2015], JSMA [Papernot et al., 2016], DeepFool [Moosavi-Dezfooli et al., 2016], PGD [Kurakin et al., 2016, Madry et al., 2018], C&W [Carlini and Wagner, 2016], EAD [Chen et al., 2017] and SparseFool [Modas et al., 2019]. Nowadays, the two most important ones are PGD with a random starting point [Madry et al., 2018] and C&W [Carlini and Wagner, 2016]. They are usually considered the state of the art for $L_\infty$ (PGD) and $L_2$ (CW). The other ones are either much weaker (FGSM, DeepFool) or minimize other norms, e.g. $L_0$ (JSMA, SparseFool) or $L_1$ (EAD). More recently, there have been some improvements to PGD that aim at making it more effective and/or more query-efficient by changing its update rule to Adam [Uesato et al., 2018] or momentum [Dong et al., 2018].

## 3   Attack algorithm

Our attacks are inspired by the decision-based boundary attack [Brendel et al., 2018] but use gradients to estimate the local boundary between adversarial and non-adversarial inputs. We will refer to this

boundary as the *adversarial boundary* for the rest of this manuscript. In a nutshell, the attack starts from an adversarial input $\tilde{\boldsymbol{x}}^0$ (which may be far away from the clean sample) and then follows the adversarial boundary towards the clean input $\boldsymbol{x}$, see Figure 1 (middle). To compute the optimal step in each iteration, Figure 1 (right), we solve a quadratic trust region optimization problem. The goal of this optimization problem is to find a step $\boldsymbol{\delta}^k$ such that (1) the updated perturbation $\tilde{\boldsymbol{x}}^k = \tilde{\boldsymbol{x}}^{k-1} + \boldsymbol{\delta}^k$ has minimal $L_p$ distance to the clean input $\boldsymbol{x}$, (2) the size $\left\|\boldsymbol{\delta}^k\right\|_2^2$ of the step is smaller than a given trust region radius $r$, (3) the updated perturbation stays within the box-constraints of the valid input value range (e.g. $[0, 1]$ or $[0, 255]$ for input) and (4) the updated perturbation $\tilde{\boldsymbol{x}}^k$ is approximately placed on the adversarial boundary.

**Optimization problem**    In mathematical terms, this optimization problem can be phrased as

$$\min_{\boldsymbol{\delta}} \left\| \boldsymbol{x} - \tilde{\boldsymbol{x}}^{k-1} - \boldsymbol{\delta}^k \right\|_p \quad \text{s.t.} \quad 0 \leq \tilde{\boldsymbol{x}}^{k-1} + \boldsymbol{\delta}^k \leq 1 \quad \wedge \quad \boldsymbol{b}^{k\top} \boldsymbol{\delta}^k = c^k \quad \wedge \quad \left\| \boldsymbol{\delta}^k \right\|_2^2 \leq r, \quad (1)$$

where $\|.\|_p$ denotes the $L_p$ norm and $\boldsymbol{b}^k$ denotes the estimate of the normal vector of the local boundary (see Figure 1) around $\tilde{\boldsymbol{x}}^{k-1}$ (see below for details). The problem can be solved for $p = 0, 1, \infty$ with off-the-shelf solvers like ECOS [Domahidi et al., 2013] or SCS [O'Donoghue et al., 2016] but the runtime of these solvers as well as their numerical instabilities in high dimensions prohibits their use in practice. We therefore derived efficient iterative algorithms based on the dual of (1) to solve Eq. (1) for $L_0, L_1, L_2$ and $L_\infty$. The additional optimization step (1) has little impact on the runtime of our attack compared to standard iterative gradient-based attacks like PGD. We report the details of the derivation and the resulting algorithms in the supplementary materials.

**Adversarial criterion**    Our attacks move along the adversarial boundary to minimize the distance to the clean input. We assume that this boundary can be defined by a differentiable equality constraint $\mathrm{adv}(\tilde{\boldsymbol{x}}) = 0$, i.e. the manifold that defines the boundary is given by the set of inputs $\{\tilde{\boldsymbol{x}} \mid \mathrm{adv}(\tilde{\boldsymbol{x}}) = 0\}$. No other assumptions about the adversarial boundary are being made. Common choices for $\mathrm{adv}(.)$ are targeted or untargeted adversarials, defined by perturbations that switch the model prediction from the ground-truth label $y$ to either a specified target label $t$ (targeted scenario) or any other label $t \neq y$ (untargeted scenario). More precisely, let $\boldsymbol{m}(\tilde{\boldsymbol{x}}) \in \mathbb{R}^C$ be the class-conditional log-probabilities predicted by model $\boldsymbol{m}(.)$ on the input $\tilde{\boldsymbol{x}}$. Then $\mathrm{adv}(\tilde{\boldsymbol{x}}) = \boldsymbol{m}(\tilde{\boldsymbol{x}})_y - \boldsymbol{m}(\tilde{\boldsymbol{x}})_t$ is the criterion for targeted adversarials and $\mathrm{adv}(\tilde{\boldsymbol{x}}) = \min_{t, t \neq y}(\boldsymbol{m}_y(\tilde{\boldsymbol{x}}) - \boldsymbol{m}_t(\tilde{\boldsymbol{x}}))$ for untargeted adversarials.

The direction of the boundary $\boldsymbol{b}^k$ in step $k$ at point $\tilde{\boldsymbol{x}}^{k-1}$ is defined as the derivative of $\mathrm{adv}(.)$,

$$\boldsymbol{b}^k = \nabla_{\tilde{\boldsymbol{x}}^{k-1}} \mathrm{adv}(\tilde{\boldsymbol{x}}^{k-1}). \tag{2}$$

Hence, any step $\boldsymbol{\delta}^k$ for which $\boldsymbol{b}^{k\top} \boldsymbol{\delta}^k = \mathrm{adv}(\tilde{\boldsymbol{x}}^{k-1})$ will move the perturbation $\tilde{\boldsymbol{x}}^k = \tilde{\boldsymbol{x}}^{k-1} + \boldsymbol{\delta}^k$ onto the adversarial boundary (if the linearity assumption holds exactly). In Eq. (1), we defined $c^k \equiv \mathrm{adv}(\tilde{\boldsymbol{x}}^{k-1})$ for brevity. Finally, we note that in the targeted and untargeted scenarios, we compute gradients for the same loss found to be most effective in Carlini and Wagner [2016]. In our case, this loss is naturally derived from a geometric perspective of the adversarial boundary.

**Starting point**    The algorithm always starts from a point $\tilde{\boldsymbol{x}}^0$ that is typically far away from the clean image and lies in the adversarial region. There are several straight-forward ways to find such starting points, e.g. by (1) sampling random noise inputs, (2) choosing a real sample that is part of the adversarial region (e.g. is classified as a given target class) or (3) choosing the output of another adversarial attack.

In all experiments presented in this paper, we choose the starting point as the closest sample (in terms of the $L_2$ norm) to the clean input which was classified differently (in untargeted settings) or classified as the desired target class (in targeted settings) by the given model. After finding a suitable starting point, we perform a binary search with a maximum of 10 steps between the clean input and the starting point to find the adversarial boundary. From this point, we perform an iterative descent along the boundary towards the clean input. Algorithm 1 provides a compact summary of the attack procedure.

## 4    Methods

We extensively compare the proposed attack against current state-of-the art attacks in a range of different scenarios. This includes six different models (varying in model architecture, defense

**Algorithm 1:** Schematic of our attacks.

**Data:** clean input $x$, differentiable adversarial criterion $\mathrm{adv}(.)$, adversarial starting point $\tilde{x}^0$
**Result:** adversarial example $\tilde{x}$ such that the distance $d(x, \tilde{x}^k) = \left\| x - \tilde{x}^k \right\|_p$ is minimized
**begin**
    $k \longleftarrow 0$
    $b^0 \longleftarrow \mathbf{0}$
    if no $\tilde{x}^0$ is given: $\tilde{x}^0 \sim \mathcal{U}(0,1)$ s.t. $\tilde{x}^0$ is adversarial (or sample from adv. class)
    **while** $k < $ *maximum number of steps* **do**
        $b^k := \nabla_{\tilde{x}^{k-1}} \mathrm{adv}(\tilde{x}^{k-1})$         `// estimate local geometry of adversarial`
         `boundary`
        $c^k := \mathrm{adv}(\tilde{x}^{k-1})$         `// estimate distance to adversarial boundary`
        $\delta^k \longleftarrow$ solve optimization problem Eq. (1) for given $L_p$ norm
        $\tilde{x}^k \longleftarrow \tilde{x}^{k-1} + \delta^k$
        $k \longleftarrow k + 1$
    **end**
**end**

mechanism and data set), two different adversarial categories (targeted and untargeted) and four different metrics ($L_0$, $L_1$, $L_2$ and $L_\infty$). In addition, we perform a large-scale hyperparameter tuning for all attacks we compare against in order to be as fair as possible. The full analysis pipeline is built on top of *Foolbox* [Rauber et al., 2017].

**Attacks** We compare against several attacks which are considered to be state-of-the-art in $L_0, L_1, L_2$ and $L_\infty$:

- *Projected Gradient Descent (PGD) [Madry et al., 2018].* Iterative gradient attack that optimizes $L_\infty$ by minimizing a cross-entropy loss under a fixed $L_\infty$ norm constraint enforced in each step.

- *Projected Gradient Descent with Adam (AdamPGD) [Uesato et al., 2018].* Same as PGD but with Adam Optimiser for update steps.

- *C&W [Carlini and Wagner, 2016].* $L_2$ iterative gradient attack that relies on the Adam optimizer, a tanh-nonlinearity to respect pixel-constraints and a loss function that weighs a classification loss with the distance metric to be minimized.

- *Decoupling Direction and Norm Attack (DDN) [Rony et al., 2018].* $L_2$ iterative gradient attack pitched as a query-efficient alternative to the C&W attack that requires less hyperparameter tuning.

- *EAD [Chen et al., 2018].* Variation of C&W adapted for elastic net metrics. We run the attack with high regularisation value ($1e^{-2}$) to approach the optimal $L_1$ performance.

- *Saliency-Map Attack (JSMA) [Papernot et al., 2016].* $L_0/L_1$ attack that iterates over saliency maps to discover pixels with the highest potential to change the decision of the classifier.

- *Sparse-Fool [Modas et al., 2019].* A sparse version of DeepFool, which uses a local linear approximation of the geometry of the adversarial boundary to estimate the optimal step towards the boundary.

**Models** We test all attacks on all models regardless as to whether the models have been specifically defended against the distance metric the attacks are optimizing. The sole goal is to evaluate all attacks on a maximally broad set of different models to ensure their wide applicability. For all models, we used the official implementations of the authors as available in the *Foolbox model zoo* [Rauber et al., 2017].

- *Madry-MNIST* [Madry et al., 2018]: Adversarially trained model on MNIST. Claim: 89.62% ( $L_\infty$ perturbation $\leq 0.3$). Best third-party evaluation: 88.42% [Wang et al., 2018].

- *Madry-CIFAR* [Madry et al., 2018]: Adversarially trained model on CIFAR-10. Claim: 47.04% ($L_\infty$ perturbation $\leq 8/255$). Best third-party evaluation: 44.71% [Zheng et al., 2018].

- *Distillation* [Papernot et al., 2015]: Defense (MNIST) with increased softmax temperature. Claim: 99.06% ($L_0$ perturbation $\leq 112$). Best third-party evaluation: 3.6% [Carlini and Wagner, 2016].

- *Logitpairing* [Kannan et al., 2018]: Variant of adversarial training on downscaled ImageNET (64 x 64 pixels) using the logit vector instead of cross-entropy. Claim: 27.9% ($L_\infty$ perturbation $\leq 16/255$). Best third-party evaluation: 0.6% [Engstrom et al., 2018].

- *Kolter & Wong* [Kolter and Wong, 2017]: Provable defense that considers a convex outer approximation of the possible hidden activations within an $L_p$ ball to optimize a worst-case adversarial loss over this region. MNIST claims: 94.2% ($L_\infty$ perturbations $\leq 0.1$).

- *ResNet-50* [He et al., 2016]: Standard vanilla ResNet-50 model trained on ImageNET that reaches 50% for $L_2$ perturbations $\leq 1 \times 10^{-7}$ [Brendel et al., 2018].

**Adversarial categories**   We test all attacks in two common attack scenarios: *untargeted* and *targeted* attacks. In other words, perturbed inputs are classified as adversarials if they are classified differently from the ground-truth label (untargeted) or are classified as a given target class (targeted).

**Hyperparameter tuning**   We ran all attacks on each model/attack combination and each sample with five repetitions and a large range of potentially interesting hyperparameter settings, resulting to between one (SparseFool) and 96 (C& W) hyperparameter settings we test for each attack. In the appendix we list all hyperparameters and hyperparameter ranges for each attack.

**Evaluation**   The success of an $L_\infty$ attack is typically quantified as the *attack success rate* within a given $L_\infty$ norm ball. In other words, the attack is allowed to perturb the clean input with a maximum $L_\infty$ norm of $\epsilon$ and one measures the classification accuracy of the model on the perturbed inputs. The smaller the classification accuracy the better performed the attack. PGD [Madry et al., 2018] and AdamPGD [Uesato et al., 2018] are highly adapted to this scenario and expect $\epsilon$ as an input.

This contrasts with most $L_0, L_1$ and $L_2$ attacks like C&W [Carlini and Wagner, 2016] or SparseFool [Modas et al., 2019] which are designed to find minimal adversarial perturbations. In such scenarios, it is more natural to measure the success of an attack as the median over the adversarial perturbation sizes across all tested samples [Schott et al., 2019]. The smaller the median perturbations the better the attack.

Our attacks also seek minimal adversarials and thus lend themselves to both evaluation schemes. To make the comparison to the current state-of-the-art as fair as possible, we adopt the success rate criterion on $L_\infty$ and the median perturbation distance on $L_0, L_1$ and $L_2$.

All results reported have been evaluated on 1000 validation samples. For the $L_\infty$ evaluation, we chose $\epsilon$ for each model and each attack scenario such that the best attack performance reaches roughly 50% accuracy. This makes it easier to compare the performance of different attacks (compared to thresholds at which model accuracy is close to zero or close to clean performance). In the untargeted scenario, we chose $\epsilon = 0.33, 0.15, 0.1, 0.03, 0.0015, 0.0006$ in the untargeted and $\epsilon = 0.35, 0.2, 0.15, 0.06, 0.04, 0.002$ in the targeted scenarios for *Madry-MNIST*, *Kolter & Wong*, *Distillation*, *Madry-CIFAR*, *Logitpairing* and *ResNet-50*, respectively.

## 5   Results

### 5.1   Attack success

In both targeted as well as untargeted attack scenarios, our attacks surpass the current state-of-the-art on every single model we tested, see Table 1 (untargeted) and Table 2 (targeted) (with the Logitpairing in the targeted $L_\infty$ scenario being the only exception). While the gains are small on some model/metric combinations like *Distillation* or *Madry-CIFAR* on $L_2$, we reach quite substantial gains on many others: on *Madry-MNIST*, our untargeted $L_2$ attack reaches median perturbation sizes of 1.15 compared to 3.24 for C&W. In the targeted scenario, the difference is even more pronounced (1.70 vs 4.79). On $L_\infty$, our attack further reduces the model accuracy by 0.1% to 14.0% relative to

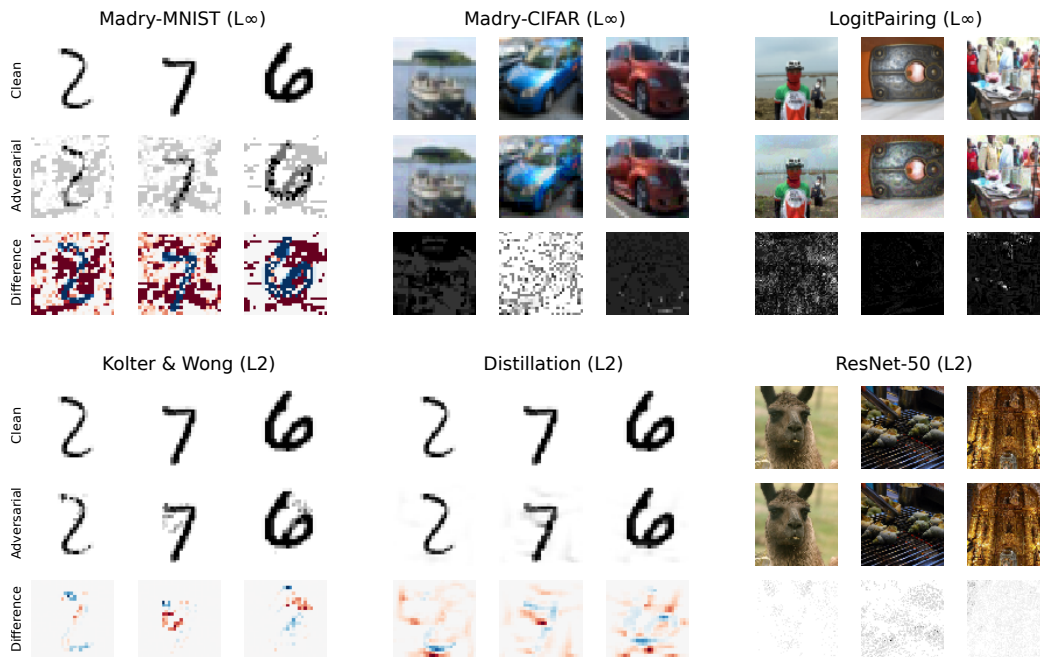

Figure 2: Randomly selected adversarial examples found by our $L_2$ and $L_\infty$ attacks for each model. The top part shows adversarial examples that minimize the $L_\infty$ norm while the bottom row shows adversarial examples that minimize the $L_2$ norm. Adversarial examples optimised with out $L_0$ and $L_1$ attacks are displayed in the appendix.

PGD. On $L_1$ and $L_0$ our gains are particularly drastic: while the SaliencyMap attack and SparseFool often fail on the defended models, our attack reaches close to 100% attack success on all models while reaching adversarials that are up to one to two orders smaller. Even the current state-of-the-art on $L_1$, EAD [Chen et al., 2018], is up to a factor six worse than our attack. Adversarial examples produced by our attacks are visualized in Figure 2 (for $L_2$ and $L_\infty$) and in the supplementary material (for $L_1$ and $L_0$).

| | MNIST | | | CIFAR-10 | ImageNet | |
|---|---|---|---|---|---|---|
| | *Madry-MNIST* | *K&W* | *Distillation* | *Madry-CIFAR* | *LP* | *ResNet-50* |
| PGD | 60.1% | 76.5% | 32.1% | 57.1% | 53.3% | 51.0% |
| AdamPGD | 53.4% | 72.5% | 31.3% | 57.1% | 53.5% | 50.2% |
| Ours-$L_\infty$ | **49.1%** | **69.5%** | **31.2%** | **57.0%** | **42.5%** | **37.0%** |
| C&W | 3.24 | 2.78 | 1.09 | 0.75 | 0.10 | 0.14 |
| DDN | 1.59 | 1.95 | 1.07 | 0.73 | 0.15 | 0.24 |
| Ours-$L_2$ | **1.15** | **1.62** | **1.07** | **0.72** | **0.09** | **0.13** |
| EAD | 0.01931 | 0.02346 | 0.00768 | 0.00285 | 0.00013 | |
| SparseFool | 0.11393 | 0.32114 | 0.48129 | 0.47687 | 0.49915 | |
| SaliencyMap | 0.04114 | 0.03730 | 0.02482 | 0.00292 | 0.00297 | |
| ours-$L_1$ | **0.00377** | **0.00707** | **0.00698** | **0.00116** | **0.00008** | |
| SparseFool | 1.00000 | 1.00000 | 1.00000 | 1.00000 | 1.00000 | |
| SaliencyMap | 0.22832 | 0.14732 | 0.08291 | 0.03483 | 0.00647 | |
| ours-$L_0$ | **0.07143** | **0.06250** | **0.00765** | **0.00228** | **0.00024** | |

Table 1: **Attack success in untargeted scenario.** Model accuracies (first block) and median adversarial perturbation distance (all other blocks) in *untargeted* attack scenarios. Smaller is better. SparseFool and SaliencyMap attacks did not always find sufficiently many adversarials to compute an overall score.

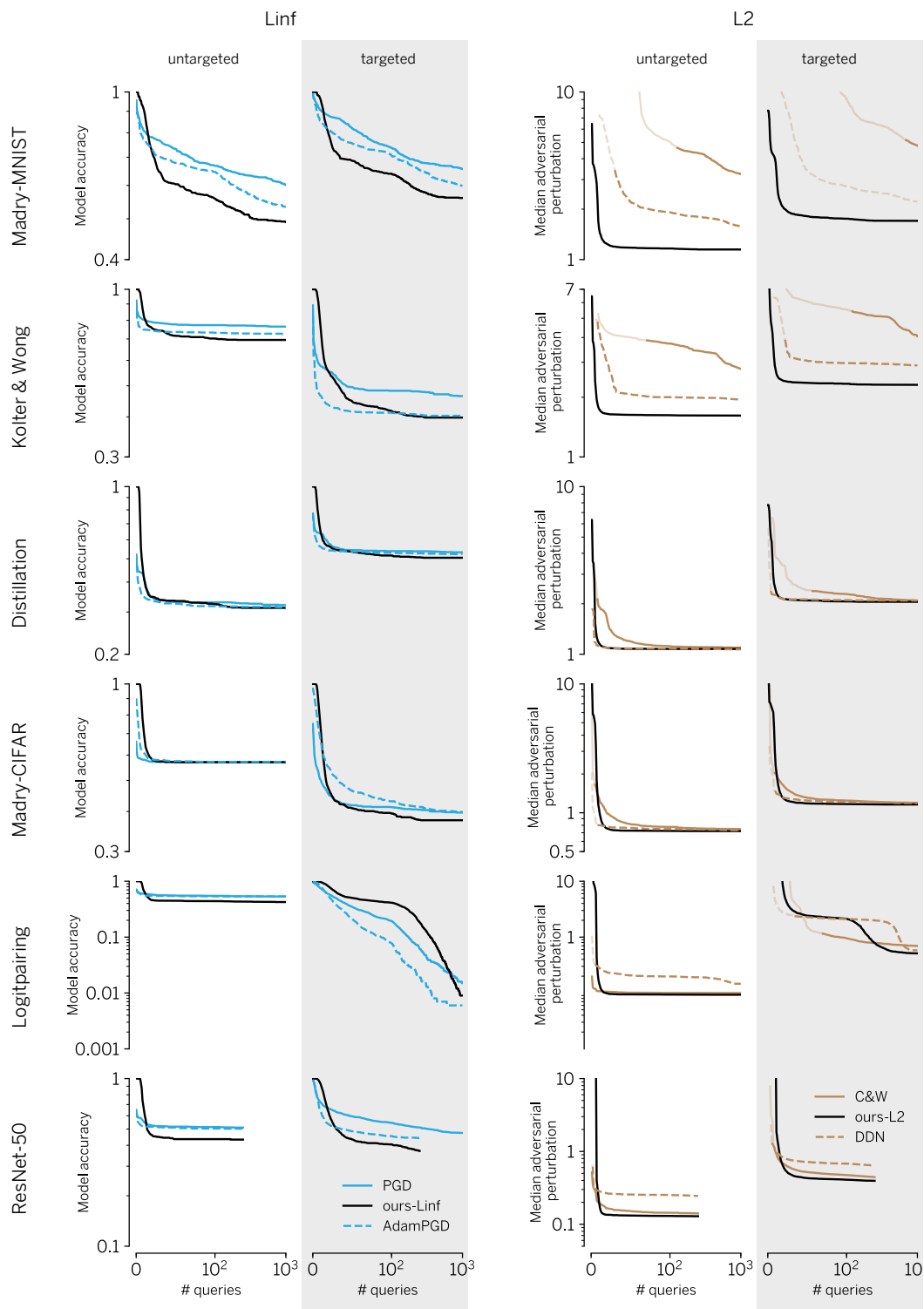

Figure 3: Query-Success curves for all model/attack combinations in the targeted and untargeted scenario for $L_2$ and $L_\infty$ metric (see supplementary information for $L_0$ and $L_1$ metric). Each curve shows the attack success either in terms of model accuracy (for $L_\infty$, left part) or median adversarial perturbation size (for $L_2$, right part) over the number of queries to the model. In both cases, lower is better. For each point on the curve, we selected the optimal hyperparameter. If no line is shown the attack success was lower than 50%. For all other points with less than 99% line is 50% transparent.

| | MNIST | | | CIFAR-10 | ImageNet | |
|---|---|---|---|---|---|---|
| | *Madry-MNIST* | *K&W* | *Distillation* | *Madry-CIFAR* | *LP* | *ResNet-50* |
| PGD | 65.6% | 46.4% | 53.2% | 39.7% | 1.5% | 47.4% |
| AdamPGD | 59.8% | 40.2% | 52.3% | 39.9% | **0.6%** | 44.1% |
| Ours-$L_\infty$ | **56.0%** | **39.8%** | **50.5%** | **37.6%** | 0.9% | **37.0%** |
| C&W | 4.79 | 4.06 | 2.09 | 1.20 | 0.70 | 0.44 |
| DDN | 2.22 | 2.89 | 2.09 | 1.19 | 0.58 | 0.64 |
| Ours-$L_2$ | **1.70** | **2.31** | **2.05** | **1.16** | **0.51** | **0.40** |
| EAD | 0.03648 | 0.04019 | 0.01808 | 0.00698 | 0.00221 | |
| SparseFool | — | — | — | — | — | — |
| SaliencyMap | 0.05740 | — | 0.03160 | 0.00872 | — | |
| ours-$L_1$ | **0.00499** | **0.00904** | **0.00925** | **0.00146** | **0.00085** | |
| SparseFool | — | — | — | — | — | — |
| SaliencyMap | 0.13074 | 0.17793 | 0.12117 | 0.04036 | — | |
| ours-$L_0$ | **0.08929** | **0.07908** | **0.01020** | **0.00293** | **0.01147** | |

Table 2: **Attack success in targeted scenario.** Model accuracies (first block) and median adversarial perturbation distance (all other blocks) in *targeted* attack scenarios. Smaller is better. SparseFool and SaliencyMap attacks did not always find sufficiently many adversarials to compute an overall score.

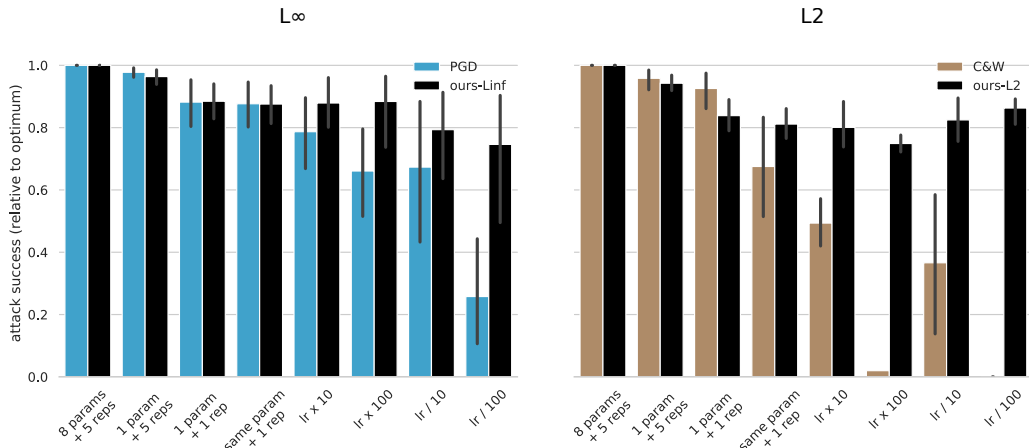

Figure 4: Sensitivity of our method to the number of repetitions and suboptimal hyperparameters.

## 5.2 Query efficiency

On $L_2$, our attack is significantly more query efficient than C&W and at least on par with DDN, see the query-distortion curves in Figure 3. Each curve represents the maximal attack success (either in terms of model accuracy or median perturbation size) as a function of query budget. For each query (i.e. each point of the curve) and each model, we select the optimal hyperparameter. This ensures that the we tease out how good each attack can perform in limited-query scenarios. We find that our $L_2$ attack generally requires only about 10 to 20 queries to get close to convergence while C&W often needs several hundred iterations. Our attack performs particularly well on adversarially trained models like Madry-MNIST.

Similarly, our $L_\infty$ attack generally surpasses PGD and AdamPGD in terms of attack success after around 10 queries. The first few queries are typically required by our attack to find a suitable initial point on the adversarial boundary. This gives PGD a slight advantage at the very beginning.

## 5.3 Hyperparameter robustness

In Figure 4, we show the results of an ablation study on $L_2$ and $L_\infty$. In the full case (*8 params + 5 reps*), we run all our attacks against C&W as well as PGD with all hyperparameter values and with five repetitions for 1000 steps on each sample and model. We then choose the smallest adversarial input across all hyperparameter values and all repetitions. This is the baseline we compare all ablations against. The results are as follows:

- Like PGD or C&W, our attacks experience only a 4% performance drop if a single hyperparameter is used instead of eight.

- Our attacks experience around 15% - 19% drop in performance for a single hyperparameter and only one instead of five repetitions, similar to PGD and C&W.

- We can even choose the same trust region hyperparameter across all models with no further drop in performance. C&W, in comparison, experiences a further 16% drop in performance, meaning it is more sensitive to per-model hyperparameter tuning.

- Our attack is extremely insensitive to suboptimal hyperparameter tuning: changing the optimal trust region two orders of magnitude up or down changes performance by less than 15%. In comparison, just one order of magnitude deteriorates C&W performance by almost 50%. Larger deviations from the optimal learning rate disarm C&W completely. PGD is less sensitive than C&W but still experiences large drops if the learning rate gets too small.

# 6 Discussion & Conclusion

An important obstacle slowing down the search for robust machine learning models is the lack of reliable evaluation tools: out of roughly two hundred defenses proposed and evaluated in the literature, less than a handful are widely accepted as being effective. A more reliable evaluation of adversarial robustness has the potential to more clearly distinguish effective defenses from ineffective ones, thus providing more signal and thereby accelerating progress towards robust models.

In this paper, we introduced a novel class of gradient-based attacks that outperforms the current state-of-the-art in terms of attack success, query efficiency and reliability on $L_0, L_1, L_2$ and $L_\infty$. By moving along the adversarial boundary, our attacks stay in a region with fairly reliable gradient information. Other methods like C&W which move through regions far away from the boundary might get stuck due to obfuscated gradients, a common issue for robustness evaluation [Athalye et al., 2018b].

Further extensions to other metrics (e.g. elastic net) are possible as long as the optimization problem Eq. (1) can be solved efficiently. Extensions to other adversarial criteria are trivial as long as the boundary between the adversarial and the non-adversarial region can be described by a differentiable equality constraint. This makes the attack more suitable to scenarios other than targeted or untargeted classification tasks.

Taken together, our methods set a new standard for adversarial attacks that is useful for practitioners and researchers alike to find more robust machine learning models.

### Acknowledgments

This work has been funded, in part, by the German Federal Ministry of Education and Research (BMBF) through the Bernstein Computational Neuroscience Program Tübingen (FKZ: 01GQ1002) as well as the German Research Foundation (DFG CRC 1233 on "Robust Vision") and the Tübingen AI Center (FKZ: 01IS18039A). The authors thank the International Max Planck Research School for Intelligent Systems (IMPRS-IS) for supporting J.R., M.K. and I.U.; J.R. acknowledges support by the Bosch Forschungsstiftung (Stifterverband, T113/30057/17); M.B. acknowledges support by the Centre for Integrative Neuroscience Tübingen (EXC 307); W.B. and M.B. were supported by the Intelligence Advanced Research Projects Activity (IARPA) via Department of Interior/Interior Business Center (DoI/IBC) contract number D16PC00003. The U.S. Government is authorized to reproduce and distribute reprints for Governmental purposes notwithstanding any copyright annotation thereon. Disclaimer: The views and conclusions contained herein are those of the authors and should not be interpreted as necessarily representing the official policies or endorsements, either expressed or implied, of IARPA, DoI/IBC, or the U.S. Government.

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
