[Supplementary Material]

# Appendix

## A    Query-Distortion curves for $L_0$ and $L_1$

Figure 1: Query-Success curves for all model/attack combinations in the targeted and untargeted scenario for $L_0$ and $L_1$ metric. Each curve shows the attack success in terms of the median adversarial perturbation size over the number of queries to the model. Lower is better. For each point on the curve, we selected the optimal hyperparameter. IIf no line is shown the attack success was lower than 50%. For all other points with less than 99% line is 50% transparent.

# B   Adversarial examples

Figure 2: Randomly selected adversarial examples found by our $L_0$ and $L_1$ attacks for each model.

# C   Solving the trust-region optimisation problem

Consider the trust-region optimisation problem defined in equation (1) of the main text for an $L_p$ metric,

$$\min_{\boldsymbol{\delta}} \|\boldsymbol{x} - \tilde{\boldsymbol{x}} - \boldsymbol{\delta}\|_p^p \quad \text{s.t.} \quad \|\boldsymbol{\delta}\|_2^2 \leq r \quad \wedge \quad \boldsymbol{b}^\top \boldsymbol{\delta} = c \quad \wedge \quad u \leq \tilde{\boldsymbol{x}} + \boldsymbol{\delta} \leq \ell \tag{1}$$

where $[u, \ell]$ is the valid interval for pixel values, $r$ is the trust region and $\boldsymbol{b}$ denotes the normal vector of the adversarial boundary. The Lagrangian of this optimisation problem is given by

$$\Lambda(\boldsymbol{\delta}, \mu, \lambda) = \|\boldsymbol{x} - \tilde{\boldsymbol{x}} - \boldsymbol{\delta}\|_p^p + \lambda(\boldsymbol{b}^\top \boldsymbol{\delta} - c) + \mu(\|\boldsymbol{\delta}\|_2^2 - r) \quad \text{s.t.} \quad \mu \geq 0 \quad \wedge \quad u \leq \tilde{\boldsymbol{x}} + \boldsymbol{\delta} \leq \ell. \tag{2}$$

The Lagrange dual function is then

$$g(\lambda, \mu) = \inf_{\boldsymbol{\delta}} \Lambda(\boldsymbol{\delta}, \mu, \lambda) \quad \text{s.t.} \quad u \leq \tilde{\boldsymbol{x}} + \boldsymbol{\delta} \leq \ell. \tag{3}$$

The dual problem is thus

$$\max_{\lambda, \mu \geq 0} g(\lambda, \mu) = \max_{\lambda, \mu \geq 0} \left[ \inf_{\boldsymbol{\delta}} \Lambda(\boldsymbol{\delta}, \mu, \lambda) \quad \text{s.t.} \quad u \leq \tilde{\boldsymbol{x}} + \boldsymbol{\delta} \leq \ell \right]. \tag{4}$$

Solving $\inf_{\boldsymbol{\delta}} \Lambda(\boldsymbol{\delta}, \mu, \lambda)$   s.t.   $u \leq \tilde{\boldsymbol{x}} + \boldsymbol{\delta} \leq \ell$ is straight-forward for all $L_p$ norms by combining principles from proximal operator theory with box-constraints (see below). This leaves us with optimising the dual problem over $\lambda$ and $\mu$ which we perform with a custom Numba

---

**Algorithm 1:** Overview over the trust-region solver for a given $L_p$ norm.

**Data:** clean image $\boldsymbol{x}$, perturbed image $\tilde{\boldsymbol{x}}$, boundary $\boldsymbol{b}$, logit-difference $c$, trust region $r$
**Result:** optimal perturbation $\boldsymbol{\delta}$ minimizing (1)
**begin**
    $\mu_0, \lambda_0 \longleftarrow 0, 0$
    **while** *not converged* **do**
        $g(\lambda_k, \mu_k) \longleftarrow \inf_{\boldsymbol{\delta}} \Lambda(\boldsymbol{\delta}, \mu_k, \lambda_k)$    s.t.    $u \leq \tilde{\boldsymbol{x}} + \boldsymbol{\delta} \leq \ell$
        $\nabla g(\lambda_k, \mu_k) \longleftarrow \nabla \inf_{\boldsymbol{\delta}} \Lambda(\boldsymbol{\delta}, \mu_k, \lambda_k)$    s.t.    $u \leq \tilde{\boldsymbol{x}} + \boldsymbol{\delta} \leq \ell$
        $\mu_{k+1}, \lambda_{k+1} \longleftarrow \text{BFGS-B}(g(\lambda_k, \mu_k), \nabla g(\lambda_k, \mu_k))$
    **end**
    $\boldsymbol{\delta}^* \leftarrow \text{arginf}_{\boldsymbol{\delta}} \Lambda(\boldsymbol{\delta}, \mu_k, \lambda_k)$    s.t.    $u \leq \tilde{\boldsymbol{x}} + \boldsymbol{\delta} \leq \ell$
**end**

---

implementation of BFGS-B (for $L_1, L_2$ and $L_\infty$) or the Nelder-Mead algorithm (for $L_0$). Since we are only optimising in a 2D space, the algorithm typically converges within 5 to 50 steps. The full algorithm is displayed in algorithm 1.

In the next subsections we show how we solve the inner optimisation problem $\inf_{\boldsymbol{\delta}} \Lambda(\boldsymbol{\delta}, \mu_k, \lambda_k)$ for the different $L_p$ norms described in this paper.

## C.1    L0 optimisation

For $L_0$ the optimisation of

$$g(\lambda, \mu) = \inf_{\boldsymbol{\delta}} \Lambda(\boldsymbol{\delta}, \mu, \lambda) = \inf_{\boldsymbol{\delta}} \|\boldsymbol{x} - \tilde{\boldsymbol{x}} - \boldsymbol{\delta}\|_0 + \lambda(\boldsymbol{b}^\top \boldsymbol{\delta} - c) + \mu(\|\boldsymbol{\delta}\|_2^2 - r) \quad \text{s.t.} \quad u \leq \tilde{\boldsymbol{x}} + \boldsymbol{\delta} \leq \ell \quad (5)$$

can be performed element-wise, i.e. we only need to solve for each index $j$

$$\inf_{\delta_j} \|\boldsymbol{x}_j - \tilde{\boldsymbol{x}}_j - \delta_j\|_0 + \lambda b_j \delta_j + \mu \delta_j^2 \quad \text{s.t.} \quad \mu \geq 0 \quad \wedge \quad u \leq \tilde{\boldsymbol{x}}_j + \delta_j \leq \ell. \quad (6)$$

Solving for each index is straight-forward: either $\delta_j = x_j - \tilde{\boldsymbol{x}}_j$ or $\delta_j = P_{u,\ell}(-\lambda b_j/(2\mu))$ where $P_{u,\ell}(.)$ is a projection on the valid region for $\delta_j$ as defined by the box-constraints, depending on which one minimizes $g(\lambda, \mu)$.

## C.2    $L_1/L_2$ optimisation

For both $L_1$ and $L_2$ metrics it is straight-forward to analytically solve $\inf_{\boldsymbol{\delta}} \Lambda(\boldsymbol{\delta}, \mu, \lambda)$   s.t.   $u \leq \tilde{\boldsymbol{x}} + \boldsymbol{\delta} \leq \ell$ by first solving the unconstrained problem for each component $\delta_j$ (i.e. without box-constraints) and then projecting the solution onto the feasible region.

## C.3    $L_\infty$ optimisation

The $L_\infty$ metric is a special case in that we cannot optimise each component of $\delta_j$ individually as they are coupled through the $L_\infty$ norm. We can simplify the optimisation, however, as follows. First, we rewrite (3) as,

$$\inf_{\boldsymbol{\delta}} \epsilon + \lambda \boldsymbol{b}^\top \boldsymbol{\delta} + \mu \|\boldsymbol{\delta}\|_2^2 \quad \text{s.t.} \quad u \leq \tilde{\boldsymbol{x}} + \boldsymbol{\delta} \leq \ell \quad \wedge \quad \|\boldsymbol{x} - \tilde{\boldsymbol{x}} - \boldsymbol{\delta}\|_\infty \leq \epsilon, \quad (7)$$

for a fixed and given $\epsilon$. This problem can be simplified by merging the box-constraints with the $L_\infty$ constraint,

$$\inf_{\boldsymbol{\delta}} \epsilon + \lambda \boldsymbol{b}^\top \boldsymbol{\delta} + \mu \|\boldsymbol{\delta}\|_2^2 \quad \text{s.t.} \quad \max(u, \boldsymbol{x} - \epsilon) \leq \tilde{\boldsymbol{x}} + \boldsymbol{\delta} \leq \min(\ell, \boldsymbol{x} + \epsilon), \quad (8)$$

which can be solved in the same way as the $L_1$ and $L_2$ optimisation problems with a suitably adapted projection operator. We then minimise the dual Lagrangian over $\epsilon$ using an adapted binary-type search algorithm to find $g(\lambda, \mu)$.

# D  Hyperparameter search

For all model/attack combinations we test each attack with a range of hyperparameters in order to select the optimal hyperparameter for each model/attack combination. Note that each point on the query-distortion curves might be realized by a different hyperparameter setting. In particular, for small query budgets higher step sizes are typically more promising while for larger query budgets the step sizes should be smaller. For each attack we use the following hyperparameters and hyperparameter ranges, all other hyperparameters are set to the default of Foolbox 2.0.0. If more than one hyperparameter is subject to a hyperparameter search, the search is performed over all possible combinations of hyperparameters.

- *PGD*

  - binary search: False
  - iterations: 1000
  - stepsize: $[1e^{-6}, 1e^{-5}, 1e^{-4}, 1e^{-3}, 1e^{-2}, 1e^{-1}, 1, 2]$

- *AdamPGD*

  - binary search: False
  - iterations: 1000
  - stepsize: $[1e^{-6}, 1e^{-5}, 1e^{-4}, 1e^{-3}, 1e^{-2}, 1e^{-1}, 1, 2]$

- *C&W*

  - learning rate: $[1e^{-4}, 1e^{-3}, 3e^{-3}, 1e^{-2}, 3e^{-2}, 1e^{-1}, 3e^{-1}, 1]$
  - initial const: $[1e^{-3}, 1e^{-2}, 1e^{-1}, 1]$
  - max iterations: $[10, 50, 250]$

- *DDN*

  - steps: 1000
  - initial norm: $[0.06, 0.1, 0.3, 0.6, 1, 2, 3, 6]$

- *EAD*

  - learning rate: $[1e^{-4}, 1e^{-3}, 3e^{-3}, 1e^{-2}, 3e^{-2}, 1e^{-1}, 3e^{-1}, 1]$
  - initial const: $[1e^{-3}, 1e^{-2}, 1e^{-1}, 1]$
  - max iterations: 200

- *Saliency-Map Attack (JSMA)*

  - steps: 1000
  - num random targets: 5

- *Sparse-Fool*

  - steps: 1000

- *ours-$L_0/L_1/L_2/L_\infty$*

  - max iterations: 1000
  - lr: $[3e^{-4}, 1e^{-3}, 3e^{-2}, 1e^{-2}, 3e^{-2}, 1e^{-1}, 3e^{-1}]$