[Reviews · NeurIPS 2019]

Reviewer 1



- Authors explained a gradient-based attack which can be used as a standard task to test the robustness of machine learning models. - The paper is well-written and problem statement is clear and concise. - There are major algorithmic and empirical contributions in this paper. - At each step, they solve a constrained quadratic program through an iterative gradient-based algorithm in order to find the most promising optimization step. After each step, the adversarial example Lp distance from the clean input example get smaller. - Authors approach, compare to previous work, only needs one hyper-parameter to tune (trust region) and as long as the boundary between the adversarial and the non-adversarial region can be described by a differentiable equality constraint, it is straightforward to extend it to other norms.

Reviewer 2



The method is inspired by previous ones, but seems to be original. The paper is well written. The paper sets out to obtain a new standard for adversarial attacks. This is an active area with many methods being proposed. Obtaining a general purpose method has been elusive. Even if the proposed method beats the state of the art, it could be quickly superseded by others. The general idea of following the boundary is interesting and further developments along these lines might take place

Reviewer 3



1. This seems to be straightforward extension of Brendel 2018. However, the novelty beyond that work seems to be quite minimal for a NeurIPS paper. Not sure if there is any attack performance improvement by adopting the gradient based strategy on top of Brendel 2018. 2. More importantly, the paper is not very clearly written and hence, rather difficult to understand. For example, the paper does not distinguish clearly between white-box and black-box attacks and therefore, it is difficult to understand the query efficiency aspect. The mathematical formulations often are not explained well, e.g., c used in Eq. 1 is explained much later. The experimental settings (e.g., hyperparameter descriptions) are also not explained in detail. I understand that there is shortage of space in the main paper, but at least more information should have been provided in the supplementary to help the readers. The same thing applies to the Budget aspect. As the notion of attack budget is not as straightforward as it is in the classical attack papers, apple-to-apple comparison here is a bit convoluted. The paper does not do a good job in explaining this aspect. Also, wondering how much budget (in the L \infty case) ends up getting used by the algorithm for different examples? A statistical estimate over a population of test examples could be useful. 3. One of the contributions that the authors listed in the abstract is the fact that their attacks are more reliable in the face of gradient masking, it will be interesting to understand (maybe geometrically) why this would be the case with the proposed approach. 4. Does the results in Table 1 and Table 2 include different \epsilon values as described in the text? I didn't really understand. 6. Being less sensitive to hyper parameter tuning is important, but as far as I could understand is that they see some benefit when one changes the learning rate significantly. I am not sure if this is a big deal in general as it is claimed in the paper. There are other elements such as initial random perturbations, number of optimization iterations etc. I would rather like to see some analysis on why the proposed algorithm (optimization problem) is fairly insensitive to learning rate variations. -------------------------------------------------------- I appreciate the authors' response, especially the response to "clarity and contributions" question well articulates the specific advancements beyond Brendel 2018. Based on the rebuttal, I am modifying my score to a 6. I encourage the authors to include the clarifications in the revised draft if the paper is accepted.

[Author Response · NeurIPS 2019]

We thank the three reviewers for their helpful feedback. We were happy to see that the reviewers were generally positive
about the manuscript and its contributions: *"There are major algorithmic and empirical contributions in this paper."*
(R1), *"seem to take reproducibility seriously"* (R2) and *"The paper is well written"* (R2).

**Further improvement: comparison to Adam-PGD (R1):** We now include Adam-PGD in our evaluations. It's better
than PGD but still mostly outperformed by our attack (Figure 1, a).

**Further improvement: extension to $L_0$ and $L_1$ (R1):** We extended our approach to $L_0$ and $L_1$. Here, our attack
shows even larger improvements over SOTA than on $L_\infty$ and $L_2$, see Figure 1 (b, c). The comparison to EAD is still
running but the preliminary results look similar. Full results will be included in the manuscript.

**Comparison to ECOS and SCS (R1):** Solvers like ECOS and SCS run into problems in our setting because they
solve general cone problems. They usually compute a sparse QR decomposition of a constraint qualification matrix. For
ImageNet, the matrix has a height and width of $224 \cdot 224 \cdot 3 + 1 \approx 150000$, for which even state-of-the-art commercial
QR solvers struggle with numerical problems. By making use of the special structure of the problem we need to solve
(only one equality constraint and simple box constraints), we can avoid factoring any matrices. For $L_2$ on ImageNet,
this decreases runtime from 10-20s to 2-20ms per iteration, underlining the relevance of our algorithmic contributions.

**Percentages, definition of query, runtime (R2):** The percentages reported on pg. 4 are model accuracies under attacks
bounded by the stated constraints (e.g., $L_\infty < 0.3$). A query includes one forward and backward pass of the attacked
model, yielding model decision and gradient. The runtime difference between a standard gradient attack and our attack
is $< 5\%$: the computational complexity of our attack is negligible compared to the model evaluation.

**Clarity and Contributions (R3):** R3 pointed out some clarity issues with regard to our distinction between black-box
and white-box attacks. We believe that due to this issue we failed to convey the contributions of our work to R3. The
original boundary attack is black-box as it only requires the final model decisions (classifications) to craft adversarials.
Our contribution is to adopt the high-level idea of the boundary attack for a gradient-based white-box attack. Compared
to the original boundary attack, which often needs 100000 queries to craft reasonably small adversarials (Brendel et
al. 2018, Figs 6,7), our version usually requires 10 to 1000 queries until convergence (which is why we focus on the
comparison with other white-box attacks). Note that our attack is not a simple adaption of the original boundary attack
(which does not estimate the boundary but just makes random steps). We here formulated a completely new algorithm
that is able to use the gradient information by solving a box-constraint trust-region problem. To solve this subproblem
we had to develop highly specialized algorithms (see our discussion above). In addition, we developed attacks for $L_0$,
$L_1$, $L_2$ and $L_\infty$ while the original boundary attack works only for $L_2$. Our proposed attacks also drastically differ from
all existing white-box attacks: while virtually all existing attacks start around the original image and follow the gradient
towards the closest adversarial (the attacks differ mainly in the optimizer, loss function or clipping), we here start from
a point far away from the original image and follow the boundary to minimize the adversarial distance.

**Evaluation of attack budget (R3):** Most papers evaluate attacks with a fixed budget, making it difficult to understand
how well the attack works for smaller or larger budgets. To show the full picture, we plot the success for all budgets
between 1 and 1000 queries (Fig. 3). For each budget (x-axis) we show model success when using the optimal
hyperparameters for this budget. This leads to a fairer comparison since we can't cherry pick the attack budget.

**Further improvement: Variability over samples (R3):** Below, we show for the MNIST-Madry model in the $L_2$ case
distortion curves with additional curves for individual samples (will be in the final paper for all models).

**Robustness to gradient masking (R3):** Gradient masking denotes the phenomenon where the decision landscape is
very flat around evaluated samples. However, at some point the decision has to change and naturally, here the gradients
will be even larger than without gradient masking. By walking along the decision boundary, we are exactly in this
region of maximal gradients. Finding the decision boundary is also robust since we use a simple binary search.

Figure 1: All query distortion/accuracy curves for Madry-MNIST: (a) $L_\infty$ metric, comparing our proposed attack
(black) with PGD (blue) and Adam-PGD (red). (b) $L_0$ metric, untargeted case. (c) $L_1$ metric, untargeted case. (d) $L_2$
metric, comparing our proposed attack (black) and C&W (green) with additional curves for individual target images.

[Meta-Review · NeurIPS 2019]

Good paper, accept. Please incorporate the clarifications provided in the rebuttal (e.g. extension of the work to L0 and L1 and adding an evaluation result for ADAM-PGD) in the final version of the paper.